# Why •CF$_2$H is nucleophilic but •CF$_3$ is electrophilic in reactions with heterocycles

Meng Duan [1], Qianzhen Shao[1], Qingyang Zhou[1], Phil S. Baran [2] & K. N. Houk [1] ✉

Radical substitution is a useful method to functionalize heterocycles, as in the venerable Minisci reaction. Empirically observed regiochemistries indicate that the CF$_2$H radical has a nucleophilic character similar to alkyl radicals, but the CF$_3$ radical is electrophilic. While the difference between •CH$_3$ and •CF$_3$ is well understood, the reason that one and two Fs make little difference but the third has a large effect is puzzling. DFT calculations with M06-2X both reproduce experimental selectivities and also lead to an explanation of this difference. Theoretical methods reveal how the F inductive withdrawal and conjugative donation alter radical properties, but only CF$_3$ becomes decidedly electrophilic toward heterocycles. Here, we show a simple model to explain the radical orbital energy trends and resulting nucleophilicity or electrophilicity of fluorinated radicals.

Fluorinated and trifluoromethylated molecular scaffolds have extensive applications in the fields of agrochemicals, pharmaceuticals, and as materials[1–4]. Even though there are several methodologies for the fluoroalkylation of organic substrates, methods for achieving the direct difluoromethylation of heterocycles are underdeveloped[5–10]. Over a decade ago, zinc difluoromethanesulfinate (Zn(SO$_2$CF$_2$H)$_2$, DFMS) was developed for direct difluoromethylation of heterocycles through a radical process[11]. This direct, scalable, user-friendly, and chemoselective difluoromethylation is compatible with a variety of organic substrates, including heterocycles, α,β-unsaturated enones, and aromatic thiols, finding extensive use in developing pharmaceuticals, agrochemicals, and other useful materials[12,13].

During these studies, it was observed empirically that the CF$_2$H radical behaves much like alkyl and aryl radicals when added to heterocycles[14–16]. All possess nucleophilic character, reacting at the electrophilic regions of the heterocycle. However, when a single additional fluorine atom is added, the CF$_3$ radical behaves as an electrophilic species. A comparison of CF$_2$H and CF$_3$ radical additions to varenicline and dihydroquinine demonstrates this (Fig. 1). In both cases, difluoromethylation takes place at electron-poor positions adjacent to nitrogen of the heterocycles (both at C2). This is found also with aryl and alkyl radicals[14–16]. In contrast, radical C−H trifluoromethylation occurs at the most electron-rich areas within the

heterocycles (C5 and C7). Why is CF$_3$ radical so much more electrophilic than CF$_2$H, which behaves like alkyl radicals with these heterocycles? Computational studies are conducted to explore and elucidate this pronounced difference, offering a comprehensive explanation for the distinct reactivity. We describe how the geometries and orbital energies of different alkyl and fluorinated radicals show a distinct discontinuity between the difluoromethyl and trifluoromethyl radicals.

## Results

It's well known that •CH$_3$ is planar and relatively nucleophilic, while •CF$_3$ is pyramidal and somewhat electrophilic[17,18]. The high electronegativity of F is countered by its ability to donate electrons by mixing of the F lone pair with the radical SOMO (Singly Occupied Molecular Orbital). As shown in Table 1, the IPs (Ionization Potential) of •CH$_3$ through •CF$_3$ gradually decrease with 1F or 2F substitutions, indicating dominance of lone-pair interaction. The IP then increases with 3F, suggesting electronegativity dominance. The absolute electronegativities ($\chi$) show similar values for CH$_3$, CH$_2$F, and CF$_2$H, but a considerably larger value for F. Similar trends are found in Parr's theoretical index, global electrophilicity ($\omega$)[19]. The computed SOMO energies also indicate that •CH$_3$, •CH$_2$F and •CF$_2$H are similar, but only •CF$_3$ is much more electrophilic. We show here why there is no gradual

[1]Department of Chemistry and Biochemistry, University of California, Los Angeles, CA 90095, USA. [2]Department of Chemistry, Scripps Research, La Jolla, CA 92037, USA. ✉e-mail: houk@chem.ucla.edu

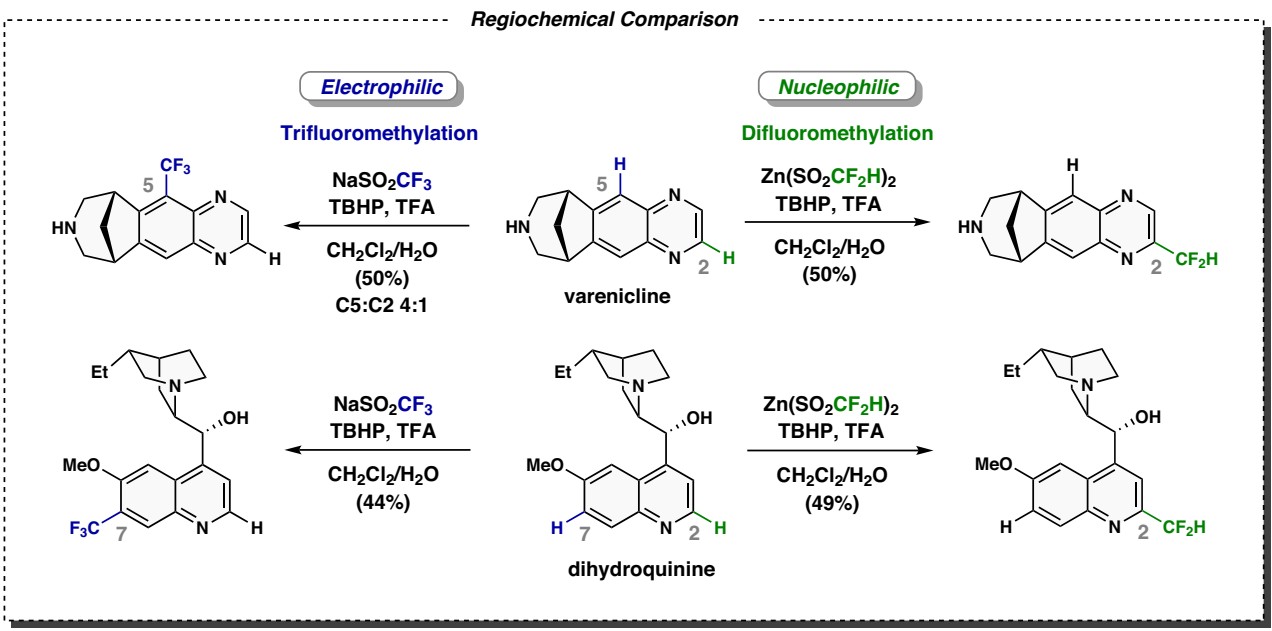

**Fig. 1 | Regiochemistries of radical difluoro- and trifluoromethylations[11].** Nucleophilic versus electrophilic radical regioselectivity.

increase in electrophilicity with the substitution of 1 and 2 Fs, but only 3Fs make the radical electrophilic.

The DFT (Density Functional Theory) calculations were conducted first to explore the differences in regioselectivity found experimentally (Fig. 2). As the reaction conditions include TFA (trifluoroacetic acid) and $H_2O$ generally in mixed $CH_2Cl_2/H_2O$ solvent, we examine $CF_2H$ and $CF_3$ radicals assuming protonated heterocycles (Fig. 2a). The transition states for attack of $CF_2H$ and $CF_3$ radicals on different positions were computed. The calculated Gibbs activation energy differences ($\Delta\Delta G_{TS}$) account for the positional selectivity. Figure 2b depicts the calculated energy of the transition states for the attack at C3, C5, or C7 as compared to the attack on C2 ($\Delta\Delta G_{TS}$). Both computed ratios (black numbers), and experimental product ratios (gray numbers in parenthesis) are presented.

As depicted in Fig. 2b, in the first case of 4-acetylpyridine **1a**, both $CF_2H$ and $CF_3$ radicals favor attack at the C2 position. The more electrophilic $CF_3$ radical shows a smaller preference for the C2 position compared to the $CF_2H$ radical (0.7 vs 1.1 kcal/mol), both experimentally and computationally. For both varenicline **1b** and dihydroquinine **1c**, the $CF_2H$ radical prefers the electron-poor position C2, while the $CF_3$ radical favors the electron-rich positions C5 (−0.8 kcal/mol) and C7 (−1.3 kcal/mol). These predicted results show the same trends as the experimental observations. In addition, a study of unprotonated heterocycles with $CF_2H$ and $CF_3$ radicals was conducted (Supplementary

Fig. 1). The predicted results with unprotonated heterocycles do not match experimentally observed selectivity, supporting our surmise that protonation occurs under these reaction conditions. These findings are in good agreement with the Baran–Blackmond rules[16].

We then focused on why $CF_3$ is so much more electrophilic than $CF_2H$, and why $CF_2H$ behaves like $CH_2F$ and $CH_3$ as well as other alkyl and aryl radicals. As shown in Fig. 3, a simple Fukui frontier molecular orbital (FMO) model considers the interaction of the SOMO with the HOMO (Highest Occupied Molecular Orbital) and LUMO (Lowest Unoccupied Molecular Orbital) of the protonated heterocycles. There are three electron (SOMO-HOMO) and one electron (SOMO-LUMO) interactions, respectively, and both are stabilizing. One of these interactions may dominate. With electrophilic protonated heterocycles, a nucleophilic radical has a high-lying SOMO, and it interacts mainly with the heterocycle LUMO. In electrophilic radicals, the SOMO energy is lower, and the SOMO-HOMO interaction also becomes important. This pattern is because protonated heterocycles are intrinsically electrophilic. A more sophisticated analysis based on open-shell calculations and a GCDA (Generalized Charge Decomposition Analysis) method is given in the SI (Supplementary Fig. 2 and 3).

Restricted open-shell DFT (RODFT) computations on geometries and FMOs of $CH_3$ through $CF_3$ are shown in Fig. 4. These RODFT calculations involve restricting all orbitals to be doubly occupied with an α and β electron, or to be vacant. Only one orbital is occupied by the radical electron, which simultaneously represents both the α-occupied orbital and the β vacant orbital. This differs from an unrestricted calculation, where α and β electrons occupy separate orbitals. The unrestricted DFT results that we have also performed are similar to RODFT but more complicated to describe, due to the presence of distinct sets of both α and β orbitals. In such cases, two different SOMO energies arise, depending on whether the SOMO is occupied or vacant.

Returning to RODFT, Fig. 4 illustrates that the SOMO energies of $CH_3$ (−2.6 eV), $CH_2F$ (−2.5 eV), $CF_2H$ (−2.8 eV), and $CF_3$ (−3.4 eV) first increase in energy and then decrease, slightly with $CF_2H$ and dramatically with $CF_3$. The percent orbital density decreases on the p orbital, by 8%, 7%, then only 3% in the series. The charge on the carbon becomes more positive along the series. Inspection of geometries indicates that there is slight pyramidalization of the radical with one fluorine, which becomes more pronounced with two and three

**Table 1 | Electronic properties of methyl and fluorinated methyl radicals[19,22,40,41]**

|  | •$CH_3$ | •$CH_2F$ | •$CF_2H$ | •$CF_3$ |
|---|---|---|---|---|
| IP [a] | 9.84 | 9.04 | 8.73 | 9.25 |
| $\chi$ [b] | 5.0 | 4.4 | 4.9 | 5.6 |
| $\omega$ [c] | 1.21 | 1.08 | 1.20 | 1.67 |
| $E_{SOMO}$ [d] | −2.6 | −2.5 | −2.8 | −3.4 |

All values are in eV.
[a]Experimental Ionization Potentials (IP).
[b]Experimental Absolute Electronegativities ($\chi = (IP + EA)/2$, EA: Electron Affinity).
[c]Global Electrophilicity ($\omega$).
[d]SOMO (Singly Occupied Molecular Orbital) energy by RODFT (Restricted Open-shell Density Functional Theory).

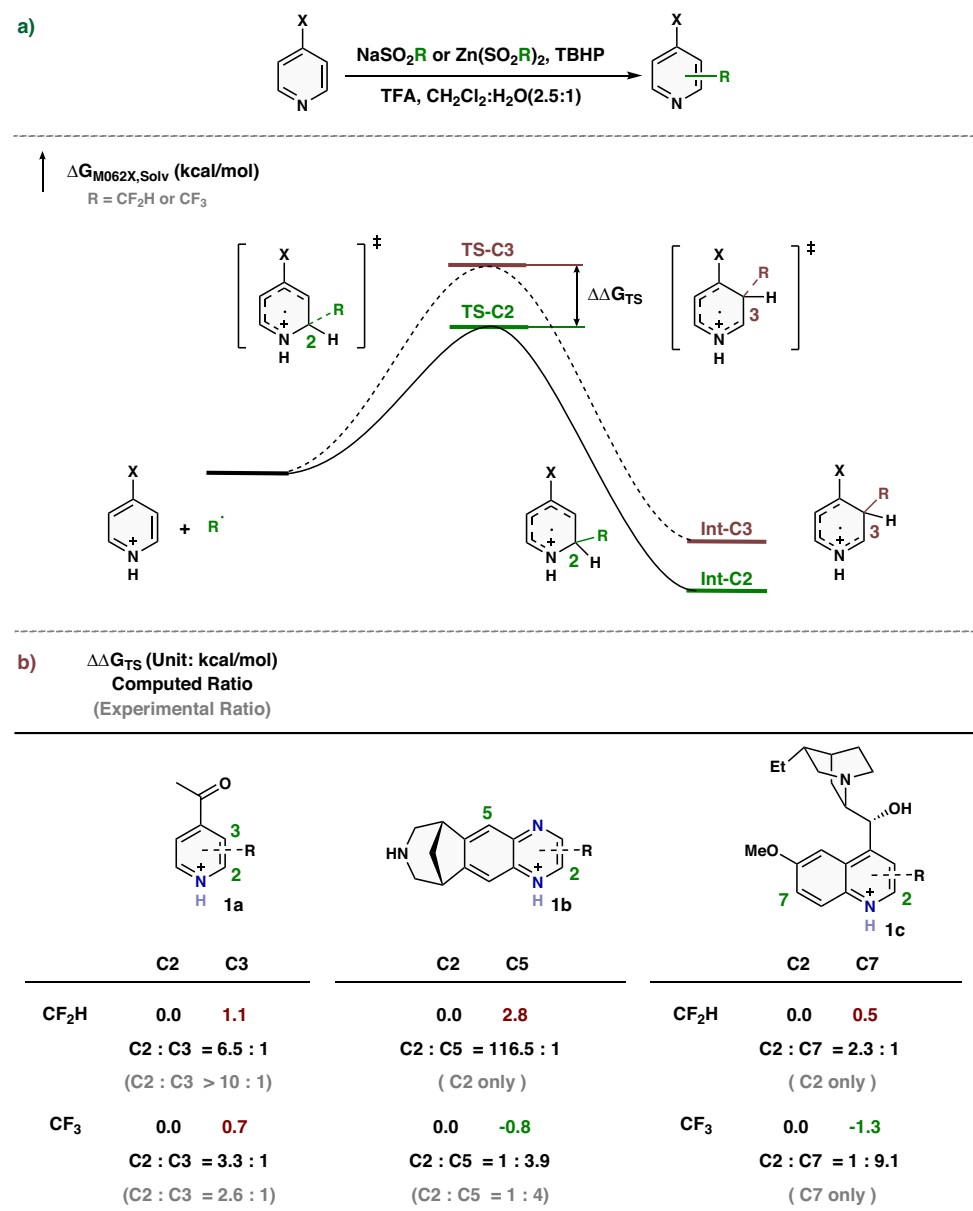

**Fig. 2 | Computational investigation. a** The energy profile and definition of $\Delta\Delta G_{TS}$. **b** Calculated relative Gibbs activation energies of transition states for the attack at C3, C5, or C7 as compared to the attack on C2 ($\Delta\Delta G_{TS}$). Computed ratios are given as black numbers, and experimental ratios of products are shown as gray numbers in parentheses. The results of the experiments are based on ref. [11].

fluorines (Fig. 4d). The pyramidalization is reflected in the deviation of the sum of the three angles at the radical center from 360° (the value for the planar methyl radical). These deviations from $CH_3$ through $CF_3$ are 0°, 7°, 21°, and 27°, a gradual pyramidalization not reflected in the abrupt difference in electrophilicity with 3Fs. There are only minor changes in the SOMO energies for $CH_2F$ and $CF_2H$ compared to $CH_3$, whereas a large decrease in the SOMO energy with the addition of three fluorines. This indicates $CF_2H$ is similar to $CH_3$, but $CF_3$ is different.

As noted earlier, fluorine substituents affect the radical SOMO energy in two ways, as depicted in Fig. 5[17,18]. One is inductive; the fluorine is electronegative, lowering the energy of the SOMO orbital. The second effect is conjugative; fluorine is a π-donor, which leads to an increase in SOMO energy, due to the antibonding interaction of the radical SOMO with the lower-lying F lone pair. At the same time, the fluorine lone-pair orbitals are stabilized by mixing with higher-energy radical SOMO orbitals in a bonding fashion.

To better clarify how fluorine π-type lone pair interacts with the radical SOMO, Fig. 6 presents a plot of different occupied orbitals, primarily p orbitals on fluorine, overlapping with the radical SOMO. A similar diagram is given in Bernardi et al.[20]. In the first case, the single fluorine lone pair mixes in a bonding fashion with the methyl SOMO orbital. In the second case, there are two fluorines, which form a bonding combination at −16.5 eV and an antibonding at −14.9 eV. However, only the −16.5 eV orbital has the right symmetry to mix with the radical SOMO orbital. As a result, the conjugative destabilization of the SOMO by antibonding interaction, and the orbital raising effect, is smaller with two fluorines than it is with one. And finally, with three fluorines, there are three different combinations of fluorine lone pairs. Only the one that involves bonding between all three fluorines, having the correct symmetry, can interact with the methyl SOMO. This orbital is much lower in energy and has smaller coefficients on fluorine, resulting in only a relatively minor destabilization of the SOMO.

The $CF_3$ radical also has a more pyramidal structure than the $CF_2H$ radical, decreasing the degree of overlap between radical SOMO and fluorine π-type lone pair. This effect is small compared to the small coefficient and lower energy of the all-bonding combination of F lone pairs in $CF_3$[20,21]. The π donation and σ withdrawal by F is well known and discussed earlier in the references[17,18,20]. However, the earlier studies generally presented a smooth transition of nucleophilic through electrophilic behavior in methyl through trifluoromethyl. We emphasize why $CH_3$ through $CHF_2$ changes little in nucleophilic/electrophilic behavior, and only $CF_3$ is different.

To provide a clearer explanation, we describe a simple model to explain the radical orbital energy in terms of F inductive withdrawal and conjugative donation (Fig. 7). In this model, the inductive effect of the fluorine is assumed to be approximately constant, as suggested by charges in Fig. 4c. Our calculations reveal that each subsequent fluorine substitution uniformly increases the charge on the carbon by a

consistent value (0.13). Consequently, we presume that the substitution of each additional fluorine reduces the SOMO energy by an identical amount, here assumed to be −0.8 eV (purple numbers). This competes with conjugative electron donation by F, which raises the energy of the SOMO. As noted in Fig. 7, the conjugative effect (green numbers) somewhat exceeds the inductive effect for one F (0.9 eV) but diminishes for the second (0.5 eV) and third (0.2 eV) fluorines. Therefore, the SOMO energies of $CH_2F$ and $CF_2H$ are similar to $CH_3$, due to near cancellation of conjugative donation and inductive withdrawal. These three radicals are nucleophilic. $CF_3$ is electrophilic because of triple F inductive withdrawal and very weak π donation. The counteracting inductive and resonance effect of F is also cited in the perfluoro effect in photoelectron spectroscopy[22], where, for example, perfluorination of benzene changes the HOMO energies (the first IP) very little.

Finally, we focus on the differences in regioselectivity between $CF_2H$ and $CF_3$ radical additions. Following the earlier discussion (Fig. 3), for the nucleophilic $CF_2H$ radical, the radical SOMO mainly interacts with the protonated heterocycle LUMO. As a result, the SOMO-LUMO interaction will be the most decisive interaction for $CF_2H$ radical addition, and the LUMO coefficients of the protonated heterocycle dominate the regioselectivity. As presented in Fig. 8, the

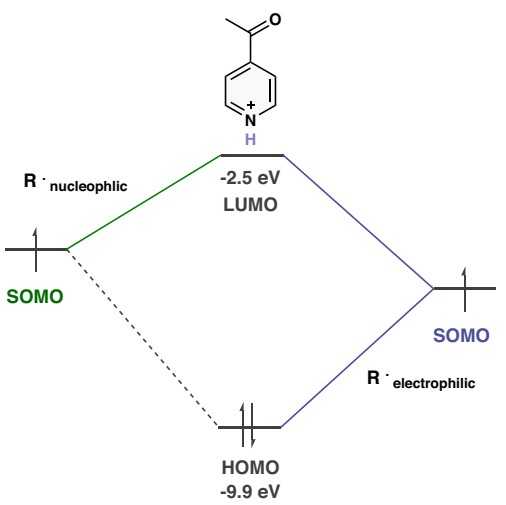

**Fig. 3 | Frontier molecular orbital model.** With electrophilic protonated heterocycles, the SOMO-LUMO interaction dominates with a nucleophilic radical, and both SOMO-HOMO and SOMO-LUMO interactions are important with an electrophilic radical.

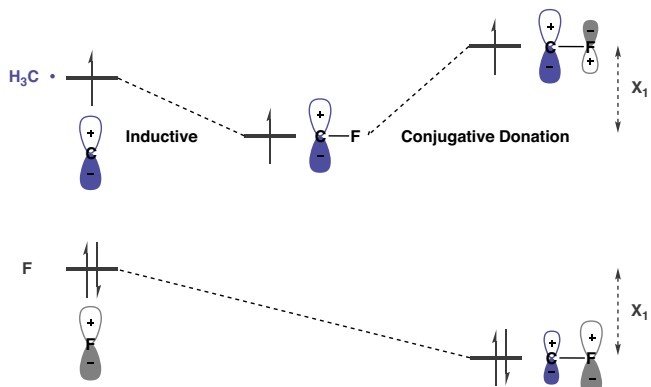

**Fig. 5 | Effects of fluorine substituents.** The orbital energy plots demonstrate the F inductive withdrawal and conjugative donation.

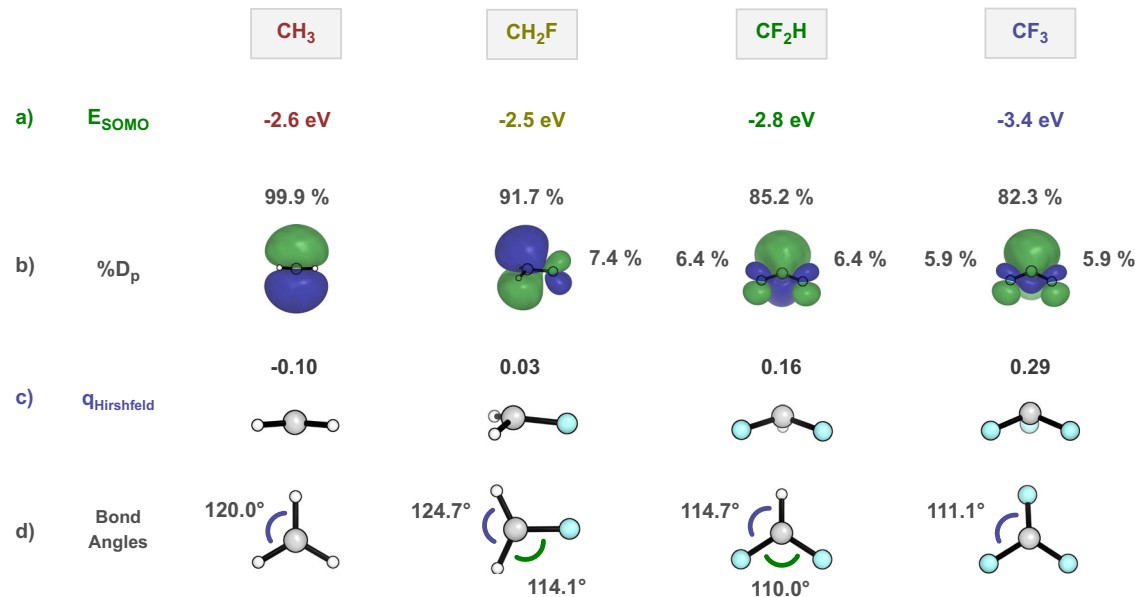

| | | CH₃ | CH₂F | CF₂H | CF₃ |
|---|---|---|---|---|---|
| a) | $E_{SOMO}$ | -2.6 eV | -2.5 eV | -2.8 eV | -3.4 eV |
| b) | %$D_p$ | 99.9 % | 91.7 %      7.4 % | 6.4 %    85.2 %    6.4 % | 5.9 %    82.3 %    5.9 % |
| c) | $q_{Hirshfeld}$ | -0.10 | 0.03 | 0.16 | 0.29 |
| d) | Bond Angles | 120.0° | 124.7°      114.1° | 114.7°    110.0° | 111.1° |

**Fig. 4 | Restricted open-shell computations. a** Restricted open-shell orbital energies of different radicals. **b** Side views of radical SOMO and percentages of orbital density at P in each SOMO (%$D_p$). **c** Side views of radicals and Hirshfeld Charges. **d** Top views of radicals and bond angles.

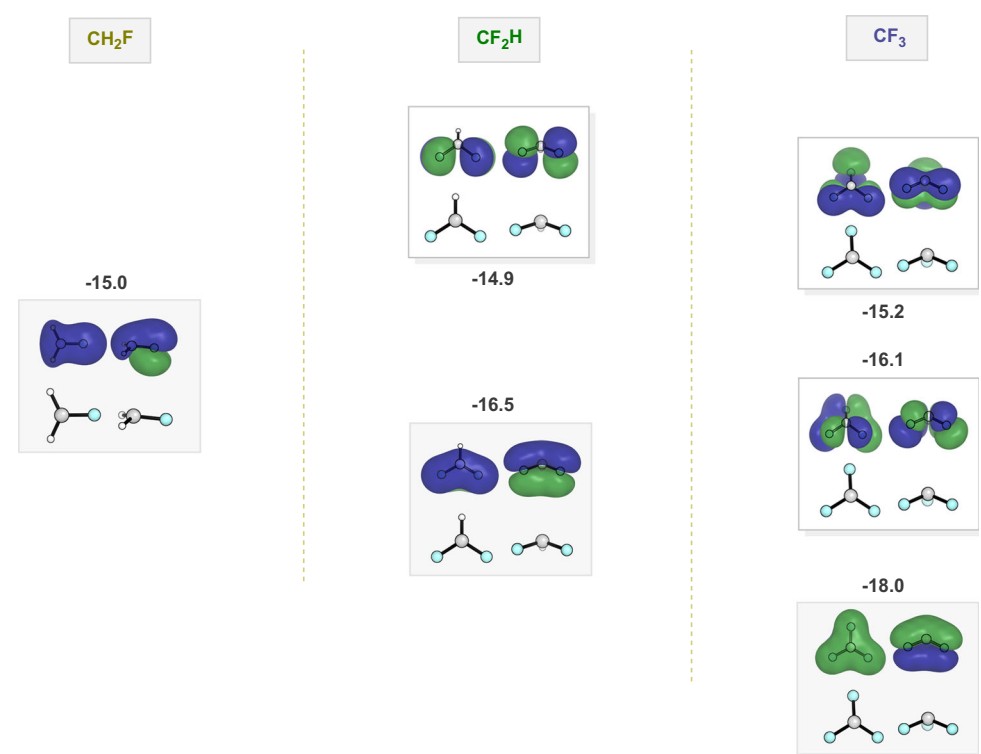

**Fig. 6 | Restricted open-shell orbital energies of F lone pairs of different radicals.** Top view and side view of each orbital are shown. Energies are in eV.

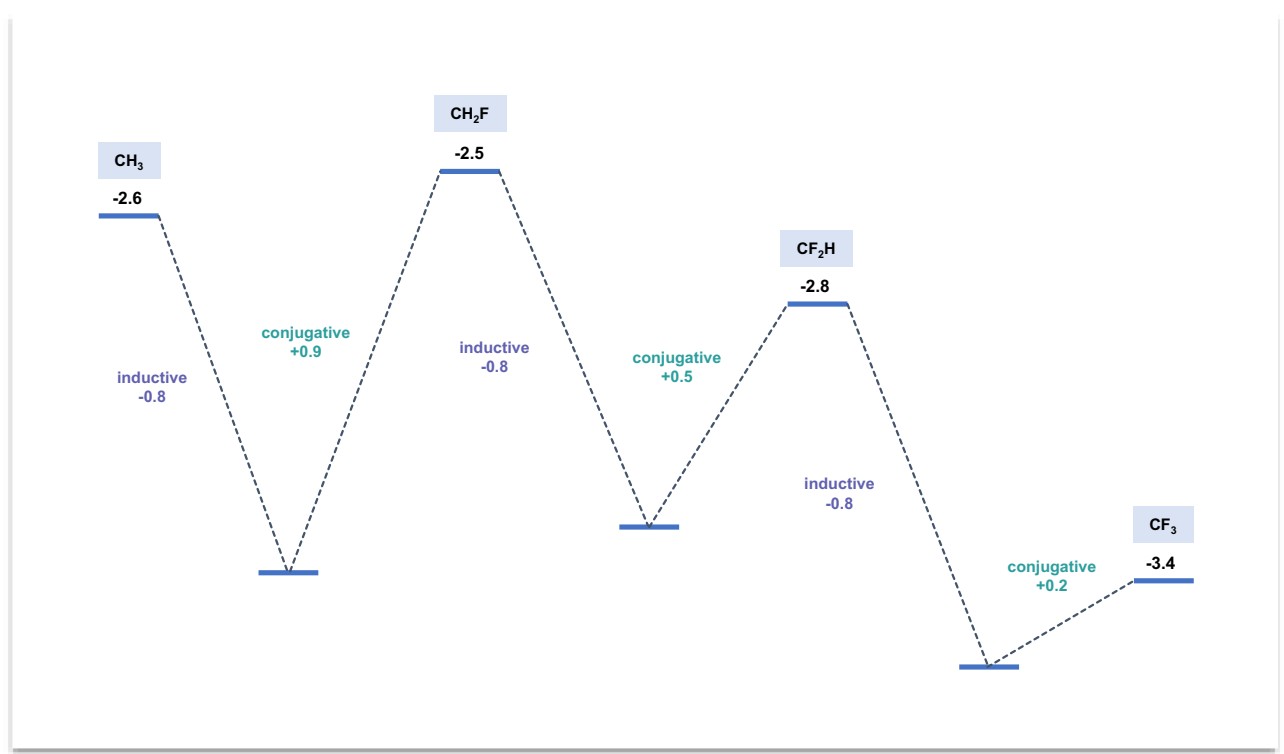

**Fig. 7 | Model to explain the radical SOMO energy in terms of F inductive withdrawal and conjugative donation.** All values are in eV. (−0.8 represents assumed constant inductive stabilization by each F; +0.9, +0.5, +0.2 represent assumed conjugative destabilization that decreases as the number of Fs increase).

order of LUMO coefficients is C2 > C3 (12.9 vs. 7.0) for the simple heterocycle 4-acetylpyridine, C2 > C5 (20.2 vs. 5.8) for varenicline, and C2 > C7 (18.7 vs. 10.3) for dihydroquinine. This matches the experimentally observed regioselectivity.

In the case of the $CF_3$ radical, the electrophilic $CF_3$ radical SOMO has a lower orbital energy, which reduces the interaction of the SOMO with the protonated heterocycle LUMO. Thus, the positional differentiation brought by the protonated heterocycle LUMO will be less pronounced. The lower radical SOMO energy leads to an increase in SOMO-HOMO interaction. Therefore, both HOMO and LUMO coefficients contribute to determining the regioselectivity of the $CF_3$ radical addition.

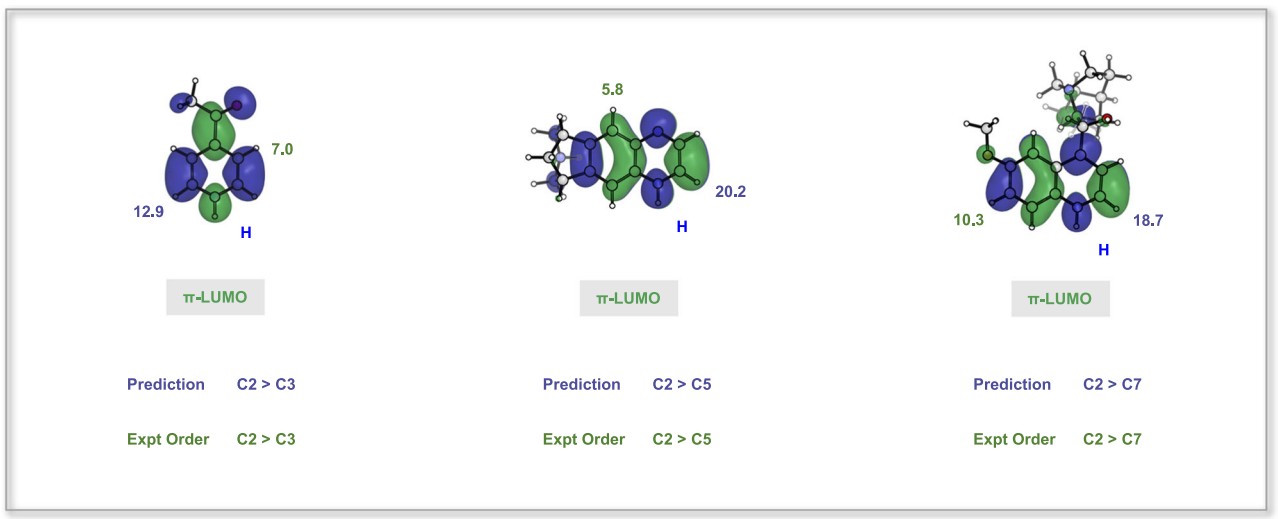

**Fig. 8 | The percentage orbital density for the protonated heterocycles, 4-acetylpyridine, varenicline, and dihydroquinine.** The predicted order for difluoromethylation is according to the LUMO densities. The experimental order is also shown.

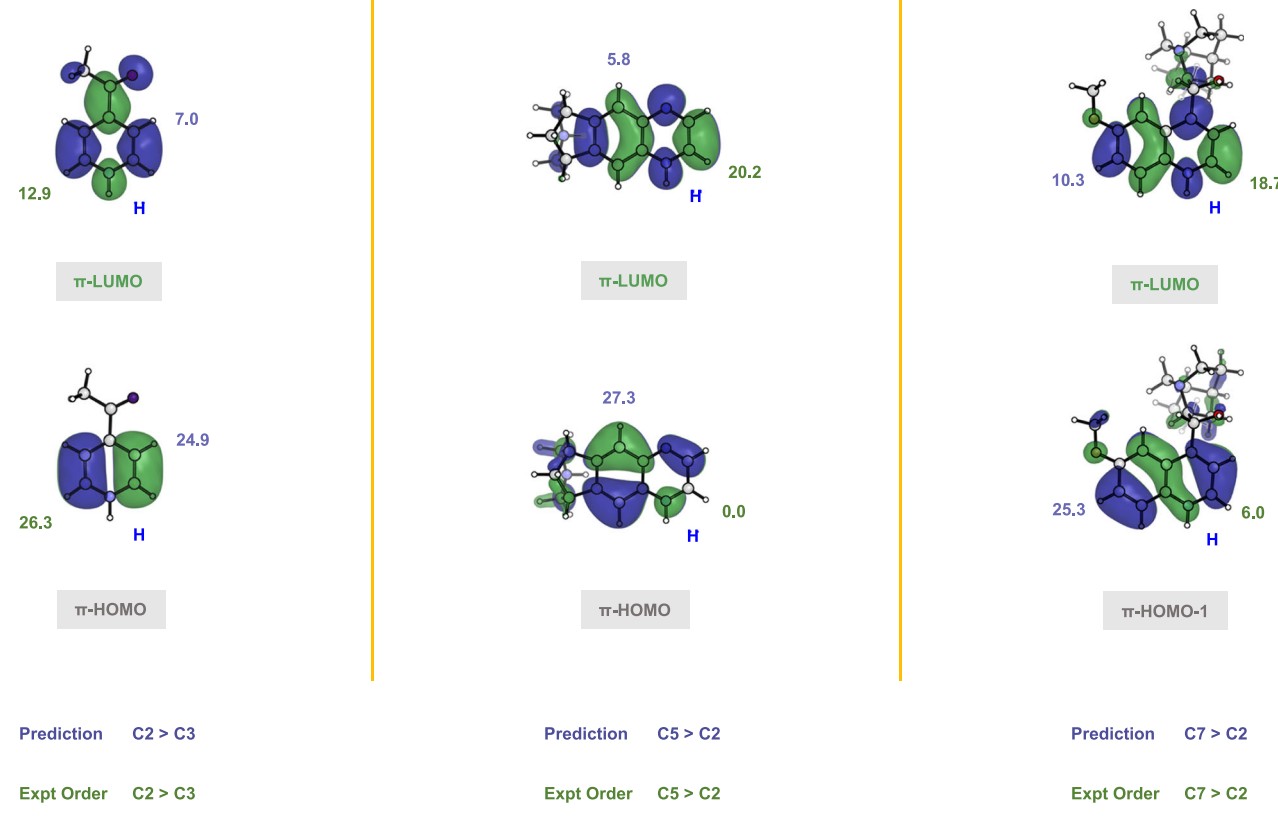

**Fig. 9 | The percentage orbital density for the protonated heterocycles, 4-acetylpyridine, varenicline, and dihydroquinine.** The predicted order for trifluoromethylation is according to the LUMO and HOMO densities. The experimental order is also shown.

As depicted in Fig. 9, for the heterocycle 4-acetylpyridine, both LUMO and HOMO coefficients are large at the C2 position. In the case of varenicline, the LUMO coefficients follow the order of C2 > C5 (20.2 vs. 5.8). By contrast, the order of HOMO coefficients is inverted (C5 » C2, 27.3 vs. 0.0). Thus, the final preference for varenicline is C5 over C2. Regarding dihydroquinine, the LUMO coefficients are ordered as C2 > C7 (18.7 vs. 10.3). Due to the node on C7 of the HOMO, the radical SOMO interacts with the dihydroquinine HOMO-1 instead of the HOMO (See Supplementary Fig. 4 for detailed explanation). The coefficients of HOMO-1 contribute to the positional differentiation.

The HOMO-1 coefficients are arranged in the order: C7 » C2 (25.3 vs. 6.0). As a result, dihydroquinine shows a preference for C7 compared to C2. This explains the experimental observations.

## Discussion

We have described DFT calculations that elucidate why $CF_2H$ is a nucleophilic radical while $CF_3$ is more electrophilic toward protonated heterocycles. To represent the F inductive withdrawal and conjugative donation, a simple model is proposed to explain the SOMO energies of $CH_3$ through $CF_3$. The analysis indicates that $CH_2F$ and $CF_2H$ have

similar SOMO energies to $CH_3$, due to the near cancellation of conjugative donation and inductive withdrawal. These are nucleophilic radicals. With three fluorines, the SOMO energy of $CF_3$ is greatly lowered, because of triple F inductive withdrawal and weak π donation. This radical is relatively electrophilic.

To demonstrate the utility of this model, regioselectivity differences between radical additions of $CF_3$ and $CF_2H$ were investigated. For the nucleophilic $CF_2H$ radical, the interaction between the radical SOMO and the heterocycle LUMO is the most decisive. Consequently, the LUMO coefficients of the heterocycle dominate the regioselectivity. In the case of more electrophilic $CF_3$ radical, the lower radical SOMO results in enhanced interaction with the heterocycle HOMO. Therefore, when dealing with electrophilic protonated heterocycles, both HOMO and LUMO coefficients have a significant effect on the regioselectivity of the $CF_3$ radical addition. The difference between •$CF_3$ and •$CF_2H$ reactivity and regioselectivity can be understood from these calculations and our FMO model.

## Methods

### Geometry optimization

All computations were performed using the Gaussian 16[23]. Geometry optimizations and frequency calculations in dichloromethane were conducted at the M06-2X/6-311 + G(d,p) level of theory with the SMD (Solvation Model Density) solvation model[24–29]. Following the geometry optimization, single-point energies and solvent effects in dichloromethane were calculated at the SMD-M06-2X/def2-QZVPP level of theory using the optimized structures[30–32]. Intrinsic Reaction Coordinate (IRC) calculations were carried out to confirm that the transition state connects the correct reactants and products.

### Frequency calculations

All structures were confirmed as stationary points on the potential energy surface (PES) and characterized as either transition states or minima through frequency calculations. Specifically, transition states were identified by the presence of a single imaginary frequency in the vibrational analysis. Minima on the PES, representing stable structures such as reactants, intermediates, or products, were characterized by the absence of imaginary frequencies. Based on the optimized structures, normal vibrational mode analysis was carried out for all stationary points to derive the thermochemical corrections for the enthalpies and free energies.

### Conformation searches

The initial phase of conformational exploration involved the use of Grimme's CREST conformer-rotamer ensemble sampling tool, version 2.10.2 with xtb version 6.3.3[33–36]. After the initial conformer generation, all the geometries below 5 kcal/mol were further refined using Gaussian 16[23]. The recalculations were conducted at the SMD-M06-2X/6-311 + G(d,p) level of theory, utilizing the Solvation Model Density (SMD) to simulate the influence of the solvent environment. This step was pivotal in accurately locating the lowest energy conformers.

### Computational analysis

The Natural Atomic Orbital (NAO) method was employed for an in-depth analysis of electronic structures[37]. This approach enables the calculation of orbital density percentages at each atom. To deepen understanding of the electronic interactions within and between molecules, the Multiwfn program was used to conduct the Generalized Charge Decomposition Analysis (GCDA) calculations[38,39].

### Difluoromethylation[11] of heterocycles standard procedures

To a solution of heterocycle (0.25 mmol, 1.0 equiv) and zinc difluoromethanesulfinate (DFMS) (200 mg, 0.50 mmol, 2.7 equiv) in dichloromethane (1.0 mL) and water (0.4 mL) at rt was added trifluoroacetic acid (20 µL, 0.25 mmol, 1.0 equiv) followed by slow addition of tert-butylhydroperoxide (70% solution in water, 0.17 mL, 1.25 mmol, 5.0 equiv) with vigorous stirring. The reaction was monitored by thin-layer chromatography until completion. For substrates that do not go to completion in 24 h, a second addition of DFMS (2.7 equiv) and tert-butylhydroperoxide (5.0 equiv) may be added to drive the reaction further. Upon consumption of the starting material, the reaction was partitioned between dichloromethane (2.0 mL) and saturated sodium bicarbonate (2.0 mL). The organic layer was separated, and the aqueous layer was extracted with dichloromethane (3 × 2.0 mL). The organic layers were dried with sodium sulfate, concentrated, and purified by column chromatography on silica gel.

### Trifluoromethylation[14] of heterocycles standard procedures

To a solution of heterocycle (0.25 mmol, 1.0 equiv) and sodium trifluoromethylsulfinate (117 mg, 0.75 mmol, 3.0 equiv) in dichloromethane (1.0 mL) and water (0.4 mL) at 0 °C was slowly added tert-butylhydroperoxide (70% solution in water, 0.17 mL, 1.25 mmol, 5.0 equiv) with vigorous stirring. The reaction was allowed to warm to room temperature and monitored by thin-layer chromatography until completion. For substrates that do not go to completion in 24 h, a second addition of sodium trifluoromethylsulfinate (3.0 equiv) and tert-butylhydroperoxide (5.0 equiv) may be added to drive the reaction towards completion. Upon consumption of the starting material, the reaction was partitioned between dichloromethane (2.0 mL) and saturated sodium bicarbonate (2.0 mL). The organic layer was separated, and the aqueous layer was extracted with dichloromethane (3 × 2.0 mL). The organic layers were dried with sodium sulfate, concentrated, and purified by column chromatography on silica gel.

## Data availability

All data generated in this study are available in the Supplementary Information or from the corresponding author upon request. Source data containing the cartesian coordinates of optimized structures are provided. Source data are provided with this paper.

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

## Acknowledgements

We are grateful to the National Science Foundation (CHE-2153972 to K.N.H.) for the financial support of this research. Calculations were performed on the Hoffman2 cluster at UCLA, and the Extreme Science and Engineering Discovery Environment (XSEDE), which is supported by the National Science Foundation (OCI-1053575 to K.N.H.).

## Author contributions

K.N.H. & P.S.B. conceived and directed the project. M.D. performed the DFT calculations. M.D., Q.S. & Q.Z. performed the data analysis. M.D., K.N.H. & P.S.B. wrote the paper with input from all other authors. All authors discussed the results and commented on the manuscript.

## Competing interests

The authors declare no competing interests.
