## [Peer Review File · Nature Communications]

Why •CF₂H is nucleophilic but •CF₃ is electrophilic in reactions with heterocyclesREVIEWER COMMENTS

Reviewer #1 (Remarks to the Author):

The authors have carried out DFT study on fluorination of heterocycles using molecular orbitals. I have serious concerns on the DFT orbitals. Besides the results are not compared with wavefunction based methods. Further, authors have not shown how their findings will augment existing fluorination chemistry.

Therefore I recommend to reject this manuscript.

Reviewer #2 (Remarks to the Author):

The paper looks into the electronic properties of CF₂H and CF₃ radicals and how they react with heteroarenes when they are difluoromethylated. The study employs DFT calculations to elucidate the differences in the behaviour of these radicals, providing valuable insights into their nucleophilic or electrophilic nature. The level of computation is appropriate. The computational results are well-supported by experimental observations, and the manuscript is generally well-written. However, some points need clarification and minor revisions.

Major Comments:

Clarity of Introduction: The introduction gives a good overview of the importance of fluorinated and trifluoromethylated molecular scaffolds, but it could use clearer links to the main topic of the study, which is the regioselectivity and electronic properties of CF₂H and CF₃ radicals in heteroarene difluoromethylation.

Figure Presentation: The figures are well-designed and contribute significantly to understanding the computational results. However, a more detailed description in the text regarding the specific information conveyed by each figure, particularly Figures 2-9, would enhance the reader's comprehension.

Chemical Structures and Terminology: In some instances, the manuscript refers to chemical structures and terminologies without providing clear definitions or explanations. A glossary or concise definitions within the text would be helpful for readers who may not be familiar with certain terms.

For example, terms like IP, EA, RODFT, SMD, SOMO, TFA, etc. should be expanded in the first instance of their use.

Minor Comments:

Formatting of Computational Methods: The section on Computational Methods could be formatted more clearly and more details added. Consider breaking down the information into subsections for geometry optimization, frequency calculations, conformation searches, and other aspects of the computational procedures to enhance readability.

Grammar and Sentence Structure: Some sentences are lengthy and could be broken down for better clarity. Additionally, proofreading for grammatical errors and consistent use of terminology is recommended.

Consistency in Figure References: Ensure consistent referencing to figures throughout the text. For example, in the discussion of Figure 2, there should be a clear reference to the specific panel or aspect being discussed.

In general, the manuscript gives a thorough computer study of the electronic properties and region-selectivity of CF₂H and CF₃ radicals. Addressing the mentioned points will further improve the clarity and coherence of the manuscript, enhancing its accessibility to a broader readership.

Reviewer #3 (Remarks to the Author):

In this work, Authors reported a work entitled “Why $\bullet\text{CF}_2\text{H}$ is a nucleophilic but $\bullet\text{CF}_3$ is a electrophilic in reactions with heterocycles” using M06-2X/6-311+G (d, p) level of theory with the SMD solvation model. From this study, they found that the F inductive withdrawal and conjugative donation alter radical properties, but only CF_3 becomes definitely electrophilic toward heterocycles. I have gone through the whole manuscript and observed that this paper is well written and presented but still this work needs to be improved. So this work may be publishable after considering the following comments in your work as given below

- 1) Authors presented the electronic properties of methyl and fluorinated methyl radicals in Table 1 but this should be discussed in the result section.
- 2) Briefly discuss and provide required expression for the determination of IP, χ , ω using theoretical method. If not possible in main manuscript then provide in supporting information.
- 3) In this work, I also found transitions states in the energy profile (see Figure 2) but there is no discussion for TS structure, imaginary frequency and IRC in Computational section. These should be discussing briefly in Computational section.
- 4) Authors used M06-2X/6-311+G(d,p) level of theory with the SMD solvation model for the geometry optimization and frequency but during single point energy calculation, Why they used def2-QZVPP basis set ? Please justify!

Reviewer #1:

Remark 1: The authors have carried out DFT study on fluorination of heterocycles using molecular orbitals. I have serious concerns on the DFT orbitals. Besides the results are not compared with wavefunction based methods. Further, authors have not shown how their findings will augment existing fluorination chemistry. Therefore, I recommend to reject this manuscript.

Our Response: Thanks for these comments. We have now done orbitals using the Hartree-Fock (HF) method (Supplementary Fig. 5). As noted by many experts, the general shapes of HF and KS orbitals, as well as relative energies for a similar series produce the same type of understanding that we have sought in this paper. Additionally, we employed the CCSD method to obtain the natural orbitals (Supplementary Fig. 6). The natural orbitals are similar to the Kohn-Sham DFT orbitals. However, it's important to note that the energies of these natural orbitals are not physically meaningful since they are not eigenfunctions of the Fock operator.

Supplementary Fig. 5 Side views of radical SOMO and restricted open-shell orbital energies of different radicals using the Hartree-Fock (HF) method.

Supplementary Fig. 6 Side views of natural orbitals using the CCSD method.

In terms of enhancing existing fluorination chemistry, our research provides a unique perspective. Previous studies, as cited, have explored the π donation and σ withdrawal effects by fluorine. These studies generally expect a smooth transition of nucleophilic through electrophilic in methyl through trifluoromethyl and find that geometry changes gradually. However, there was no recognition, nor explanation, of why CH₃ through CHF₂ change little in reactivity, and only CF₃ is different. Our

paper addresses this by developing a simple model to explain the SOMO energy trends and resulting nucleophilicity or electrophilicity of fluorinated radicals. This model gives chemists a way to understand this difference. We believe this work will be of great interest to organic, inorganic, and computational chemist readers of *Nature Communications*.

Reviewer #2:

The paper looks into the electronic properties of CF_2H and CF_3 radicals and how they react with heteroarenes when they are difluoromethylated. The study employs DFT calculations to elucidate the differences in the behaviour of these radicals, providing valuable insights into their nucleophilic or electrophilic nature. The level of computation is appropriate. The computational results are well-supported by experimental observations, and the manuscript is generally well-written. However, some points need clarification and minor revisions.

Major Comments:

Remark 1: Clarity of Introduction: The introduction gives a good overview of the importance of fluorinated and trifluoromethylated molecular scaffolds, but it could use clearer links to the main topic of the study, which is the regioselectivity and electronic properties of CF_2H and CF_3 radicals in heteroarene difluoromethylation.

Our Response: Thank you for these comments. We understand your observation that while the introduction effectively highlights the importance of fluorinated and trifluoromethylated molecular scaffolds, it lacks clear connections to the main focus of our study: the regioselectivity and electronic properties of CF_2H and CF_3 radicals in heteroarene difluoromethylation. To address this, we have revised the introduction to establish a more direct link between the general discussion of fluorinated scaffolds and our specific research topic. The changes in the manuscript have been highlighted with a yellow background.

Remark 2: Figure Presentation: The figures are well-designed and contribute significantly to understanding the computational results. However, a more detailed description in the text regarding the specific information conveyed by each figure, particularly Figures 2-9, would enhance the reader's comprehension.

Our Response: Thanks for these comments. We appreciate your recommendation to provide more detailed descriptions of the information conveyed by each figure, particularly for Figures 2-9. We agree that enhancing the textual description of these figures would significantly enhance the reader's comprehension. To address this, we have revised the manuscript to include more detailed explanations of Figures 2-9.

Remark 3: Chemical Structures and Terminology: In some instances, the manuscript refers to chemical structures and terminologies without providing clear definitions or explanations. A glossary or concise definitions within the text would be helpful for readers who may not be familiar with certain terms. For example, terms like IP, EA, RODFT, SMD, SOMO, TFA, etc. should be expanded in the first instance of their use.

Our Response: Thank you for highlighting the importance of clarity in our manuscript, especially regarding the use of specialized chemical terms. To address this, we have ensured that each term is clearly defined upon its first appearance in the text. For instance, terms such as IP (Ionization Potential), EA (Electron Affinity), RODFT (Restricted Open-shell Density Functional Theory), SMD (Solvation Model Density), SOMO (Singly Occupied Molecular Orbital), and TFA (trifluoroacetic acid) have been expanded in their respective contexts.

Minor Comments:

Remark 4: Formatting of Computational Methods: The section on Computational Methods could be formatted more clearly and more details added. Consider breaking down the information into subsections for geometry optimization, frequency calculations, conformation searches, and other aspects of the computational procedures to enhance readability.

Our Response: Thanks for these comments. To enhance clarity and readability, the Methods section has been restructured into distinct subsections, each focusing on a specific aspect of the computational procedures employed in this study.

Methods

Geometry Optimization: All computations were performed using the Gaussian 16.²³ Geometry optimizations and frequency calculations in dichloromethane were conducted at the M06-2X/6-311+G(d,p) level of theory with the SMD (Solvation Model Density) solvation model.²⁴⁻²⁹

Following the geometry optimization, single-point energies and solvent effects in dichloromethane were calculated at the SMD-M06-2X/def2-QZVPP level of theory using the optimized structures.³⁰⁻³² Intrinsic Reaction Coordinate (IRC) calculations were carried out to confirm that the transition state connects the correct reactants and products.

Frequency Calculations: All structures were confirmed as stationary points on the potential energy surface (PES) and characterized as either transition states or minima through frequency calculations. Specifically, transition states were identified by the presence of a single imaginary frequency in the vibrational analysis. Minima on the PES, representing stable structures such as reactants, intermediates, or products, were characterized by the absence of imaginary frequencies. Based on the optimized structures, normal vibrational mode analysis was carried out for all stationary points to derive the thermochemical corrections for the enthalpies and free energies.

Conformation Searches: The initial phase of conformational exploration involved the use of Grimme's CREST conformer-rotamer ensemble sampling tool, version 2.10.2 with xtb version 6.3.3.³³⁻³⁶ After the initial conformer generation, all the geometries below 5 kcal/mol were further refined using Gaussian 16.²³ The recalculations were conducted at the SMD-M06-2X/6-311+G(d,p) level of theory, utilizing the Solvation Model Density (SMD) to simulate the influence of the solvent environment. This step was pivotal in accurately locating the lowest energy conformers.

Computational Analysis: The Natural Atomic Orbital (NAO) method was employed for an in-depth analysis of electronic structures.³⁷ This approach enables the calculation of orbital density percentages at each atom. To deepen understanding of the electronic interactions within and between molecules, the Multiwfn program was used to conduct the Generalized Charge Decomposition Analysis (GCDA) calculations.³⁸⁻³⁹

Remark 5: Grammar and Sentence Structure: Some sentences are lengthy and could be broken down for better clarity. Additionally, proofreading for grammatical errors and consistent use of terminology is recommended.

Our Response: Thank you for these comments. We agree with that some sentences are overly lengthy and may benefit from being broken down for improved clarity and readability. We have carefully reviewed the entire manuscript and revised the longer sentences, dividing them into

shorter, more concise ones. Additionally, we have proofread once again to correct any grammatical errors and to ensure consistent use of terminology throughout the manuscript.

Remark 6: Consistency in Figure References: Ensure consistent referencing to figures throughout the text. For example, in the discussion of Figure 2, there should be a clear reference to the specific panel or aspect being discussed.

Our Response: Thanks for these comments. We appreciate your attention to detail and agree with your suggestion. Ensuring clear and precise references to figures is crucial for the reader's understanding and ease of following the arguments and data presented. To address this, we have thoroughly reviewed the manuscript and made the necessary revisions. We believe these changes will enhance the readability of our manuscript.

In general, the manuscript gives a thorough computer study of the electronic properties and region-selectivity of CF_2H and CF_3 radicals. Addressing the mentioned points will further improve the clarity and coherence of the manuscript, enhancing its accessibility to a broader readership.

Reviewer #3:

In this work, Authors reported a work entitled “Why $\bullet\text{CF}_2\text{H}$ is a nucleophilic but $\bullet\text{CF}_3$ is a electrophilic in reactions with heterocycles” using M06-2X/6-311+G (d, p) level of theory with the SMD solvation model. From this study, they found that the F inductive withdrawal and conjugative donation alter radical properties, but only CF_3 becomes definitely electrophilic toward heterocycles. I have gone through the whole manuscript and observed that this paper is well written and presented but still this work needs to be improved. So this work may be publishable after considering the following comments in your work as given below

Remark 1: Authors presented the electronic properties of methyl and fluorinated methyl radicals in Table 1 but this should be discussed in the result section.

Our Response: Thank you for these comments. We agree that moving this discussion to the results section will provide a clearer and more direct connection between these established electronic properties and the new findings of our study. In line with this, the data from Table 1 are now discussed in the results section.

Remark 2: Briefly discuss and provide required expression for the determination of IP, χ , ω using theoretical method. If not possible in main manuscript then provide in supporting information.

Our Response: Thanks for these comments. Detailed methodologies and expressions have been included in the supporting information.

1. Ionization Potential (IP):

Ionization potential is the energy required to remove an electron from the outermost shell of a molecule. In Density Functional Theory (DFT), it can be calculated using the following expression:

$$\text{IP} = E(\text{cation}) - E(\text{neutral})$$

where $E(\text{cation})$ represents the energy of the ionized molecule (cationic state), and $E(\text{neutral})$ is the energy of the molecule in its neutral state.

2. Absolute Electronegativity (χ):

Absolute electronegativity represents the tendency of an atom or molecule to attract electrons. In DFT, absolute electronegativity can be calculated using the Mulliken electronegativity concept with the formula:

$$\chi = (\text{IP} + \text{EA})/2$$

where IP stands for the Ionization Potential, and EA refers to the Electron Affinity. The Electron Affinity is determined by the formula $\text{EA} = E(\text{neutral}) - E(\text{anion})$, where $E(\text{anion})$ is the energy of the anionic molecule.

3. Global Electrophilicity (ω):

Global Electrophilicity quantifies the electrophilic nature of a molecule, essentially measuring its ability to accept electrons. In the DFT framework, global electrophilicity can be calculated using the formula:

$$\omega = \mu^2/(2\eta)$$

where μ is the electronic chemical potential, typically approximated as the negative of absolute electronegativity (χ), and η is the chemical hardness, calculated by $\eta = (\text{IP} - \text{EA})/2$.

Remark 3: In this work, I also found transition states in the energy profile (see Figure 2) but there is no discussion for TS structure, imaginary frequency and IRC in Computational section. These should be discussed briefly in Computational section.

Our Response: Thank you for the reminder. Discussions of TS structure, imaginary frequency and IRC are now included in the revised Computational section.

Geometry Optimization: All computations were performed using the Gaussian 16.²³ Geometry optimizations and frequency calculations in dichloromethane were conducted at the M06-2X/6-311+G(d,p) level of theory with the SMD (Solvation Model Density) solvation model.²⁴⁻²⁹ Following the geometry optimization, single-point energies and solvent effects in dichloromethane were calculated at the SMD-M06-2X/def2-QZVPP level of theory using the optimized structures.³⁰⁻³² Intrinsic Reaction Coordinate (IRC) calculations were carried out to confirm that the transition state connects the correct reactants and products.

Frequency Calculations: All structures were confirmed as stationary points on the potential energy surface (PES) and characterized as either transition states or minima through frequency calculations. Specifically, transition states were identified by the presence of a single imaginary frequency in the vibrational analysis. Minima on the PES, representing stable structures such as reactants, intermediates, or products, were characterized by the absence of imaginary frequencies. Based on the optimized structures, normal vibrational mode analysis was carried out for all stationary points to derive the thermochemical corrections for the enthalpies and free energies.

4) Authors used M06-2X/6-311+G(d,p) level of theory with the SMD solvation model for the geometry optimization and frequency but during single point energy calculation, Why they used def2-QZVPP basis set ? Please justify!

Our Response: Thanks for these comments. We chose the M06-2X/6-311+G(d,p) level of theory for geometry optimization and frequency calculations due to its accurate prediction of geometries and vibrational frequencies. When combined with the SMD solvation model, this approach yields reliable geometrical and vibrational data in solvated environments. For single-point energy calculations, we opted for the def2-QZVPP basis set. This selection was based on the ability of the def2-QZVPP basis set to provide a more comprehensive and accurate description of electron correlation effects, especially in systems with complex electronic structures. Being a quadruple-zeta

valence basis set with polarization and diffuse functions, the def2-QZVPP basis set enhances the precision of our electronic energy calculations. The combination of the M06-2X functional with the def2-QZVPP basis set for single-point energy calculations is a strategic choice aimed at enhancing the accuracy of the computed electronic energies. This approach provides a good balance between computational cost and accuracy.

Thanks very much for your consideration of our manuscript.

With best regards

K. N. Houk
Distinguished Research Professor

REVIEWERS' COMMENTS

Reviewer #2 (Remarks to the Author):

The authors have addressed all my concerns. I now recommend acceptance of the manuscript.

Reviewer #3 (Remarks to the Author):

[Editor's note: In comments to the editorial office, the reviewer remarked that their concerns were addressed, and the manuscript is now suitable for publication.]